# Post-Traumatic Stress and Coping Strategies of South African Nurses during the Second Wave of the COVID-19 Pandemic

**DOI:** 10.3390/ijerph18157919

**Published:** 2021-07-27

**Authors:** Michelle C. Engelbrecht, J. Christo Heunis, N. Gladys Kigozi

**Affiliations:** Centre for Health Systems Research & Development, Faculty of The Humanities, University of the Free State, P.O. Box 339, Bloemfontein 9300, South Africa; heunisj@ufs.ac.za (J.C.H.); kigozign@ufs.ac.za (N.G.K.)

**Keywords:** COVID-19, nurse, post-traumatic stress, coping strategies

## Abstract

Prior to the 2019 novel coronavirus (COVID-19) outbreak, the South African healthcare system was already under severe strain due to amongst others, a lack of human resources, poor governance and management, and an unequal distribution of resources among provinces and between the public and private healthcare sectors. At the center of these challenges are nurses, the backbone of the healthcare system, and the first point of call for most patients in the country. This research investigated post-traumatic stress and coping strategies of nurses during the second wave of COVID-19 in the country. A structured self-administered questionnaire captured the biographic characteristics, perceived risk factors for COVID-19, and views on infection control of 286 nurses Data were subjected to descriptive and binomial logistic regression analyses. More than four in every 10 nurses screened positive for higher levels of post-traumatic disorder (PTSD). Self-reported risk for contracting COVID-19 mainly centered on being a health worker and patients’ non-adherence to infection prevention guidelines. Unpreparedness to manage COVID-19 patients, poorer health, and avoidant coping were associated with PTSD. Nurses voiced a need for emotional support and empathy from managers. Emotional, psychological, and debriefing intervention sessions that focus on positive coping strategies to actively address stress are recommended.

## 1. Introduction

Pandemics are a unique form of disaster, resulting in adverse psychological symptoms and behavioral reactions [1]. Healthcare workers (HCWs) are at the frontline of responses to pandemics, increasing their risk of infection as well as psychological distress [2]. This was evidenced in earlier disease outbreaks like Ebola and Severe Acute Respiratory Syndrome (SARS). The World Health Organization reported that HCWs were up to 32 times more likely to contract Ebola than the general population during the 2014–2016 outbreak in West Africa [3]. During the 2003 SARS epidemic, one index patient was found to infect up to 50 HCWs [4]. A Canadian study found that, one to two years after SARS, HCWs still experienced higher levels of burnout, psychological distress and post-traumatic stress disorder (PTSD) compared to HCWs who did not care for SARS patients [5].

Prior to the 2019 novel coronavirus SARS-CoV2 (COVID-19) outbreak, the South African healthcare system was already under severe strain due to amongst others multiple disease burdens, a lack of human resources for health, poor governance and management, and an unequal distribution of resources among provinces and between the public and private healthcare sectors [6,7,8]. At the center of these challenges are nurses, the backbone of the healthcare system, and the first point of call for most patients in the country. South African nurses have thus been experiencing stress and burnout long before the outbreak of COVID-19 [9,10,11]. A combination of difficulties within the healthcare system, existing stress and burnout, in addition to the COVID-19 burden which entails working on the frontline; caring for infected patients while facing exhaustion, difficult triage decisions, separation from families, stigma, the pain of losing patients and colleagues; as well as their own risk of infection [2] place nurses in a precarious position. In this regard, the South African Minister of Health reported that as of 31 December 2020, there were 43,124 confirmed COVID-19 cases among HCWs in the public sector and 439 deaths [12]. Earlier statistics indicated that more than half of the infected HCWs were nurses [13].

While there are international studies, largely emanating from China, reporting on the psychological impact of COVID-19 on HCWs, there is limited research in this field in Africa. Studies in Wuhan, China found that frontline HCWs are at increased risk for mental health disturbances such as anxiety, depression and distress [14,15,16,17]. Similar studies in the United States of America [18,19], United Kingdom [20,21], Ireland [22], and Portugal [23] found that COVID-19 is an occupational hazard for healthcare professionals, who experienced higher levels of anxiety, stress and depressive symptoms compared to the general population. These findings are also confirmed by rapid reviews of existing literature [24,25,26,27]. One South African study conducted by the Human Sciences Research Council between April and May 2020 found that approximately 20% of HCWs were severely distressed, whilst just over half the sample had low distress. Psychological distress was significantly higher among nurses than medical practitioners and other healthcare professionals, and among public sector employees than those in the private or other sectors [28].

Given the historical and current impact of epidemics on HCWs, the COVID-19 response needs to place significant emphasis on the protection and well-being of HCWs [29]. Further, as noted by Kim et al. [19], little is known about the association between various coping mechanisms and nurses’ mental health during the COVID-19 pandemic. Such knowledge could inform the adaptation and/or development of interventions to improve and sustain the psychological well-being of frontline nurses during the COVID-19 crisis. This paper will focus on post-traumatic stress and coping strategies of nurses during the COVID-19 pandemic. A meta-analysis and systematic review of the prevalence of PTSD after infectious disease pandemics in the 21st century reported that PTSD following pandemics is a serious public health concern, particularly among HCWs [30]. Post-traumatic stress is defined as “*a psychiatric disorder that may occur in people who have experienced or witnessed a traumatic event such as a natural disaster, a serious accident, a terrorist act, war/combat, or rape or who have been threatened with death, sexual violence or serious injury*” [31]. Failure to receive treatment for PTSD can have long-term harmful effects on the individual’s ability to function socially and at work; it can have an impact on family life and personal health. As such, it is important for a public mental health response to address PTSD, especially post-pandemic and even longer term. Early detection of PTSD among vulnerable populations, including nurses, is important to improve post-pandemic mental health and recovery [30]. With this in mind, the objectives of the study are to (1) identify perceptions of risk for COVID-19 infection; (2) determine levels of post-traumatic stress experienced by nurses; (3) identify coping mechanisms used during COVID-19; (4) investigate associations between different coping strategies and post-traumatic stress; (5) describe the support needs of nurses.

## 2. Materials and Methods

### 2.1. Design and Setting

A cross-sectional survey was conducted among nurses working in all healthcare facilities in the Free State Province. South Africa has a two-tiered healthcare system comprising an under-resourced public sector and a well-resourced private sector. The Province has an estimated 221 government run primary healthcare (PHC) facilities, 31 public hospitals and 19 private hospitals [32]. Most South Africans (84%) access healthcare through government-run facilities, which are underfunded and understaffed. The public health system is tax-funded—PHC services are offered for free, while free hospital services are subject to a means test. Anyone can access public health services; however, access to private healthcare depends on the individual’s ability to pay for these services. Therefore, the wealthiest 20% of the population uses the private system and is far better served [33].

### 2.2. Sample and Data Collection

The study population was all categories of nurses (18 years and older)—professional, enrolled and nursing assistants—working in public and private healthcare facilities (both hospitals and PHC facilities), as well as nurse training institutions in the Free State province. In 2019, the Free State had a total of 5209 nurses—2175 professional nurses, 996 enrolled nurses and 2038 nursing assistants [34].

The survey was open to all nurses in the Free State and was advertised on multiple platforms, including mainstream media (radio) and social medium platforms (Facebook). The social media campaigns reached 19,272 people. The provincial health department also advertised the survey on official communication channels and WhatsApp groups. In addition, we contacted all hospital Chief Executive Officers (CEOs) via e-mail and followed up with telephone calls to nursing services managers and PHC facility managers to inform them about the study and encourage them to ask nurses in their facilities to participate. The survey was open from 5 December 2020 to 3 March 2021. During this period, South Africa was experiencing a second wave of COVID-19 infections. The South African National Institute for Communicable Diseases [35] defined the second wave as, “a new wave lasting one or more days, commencing after the ‘end of the first wave’”. It furthermore refers to an occurrence, after the previous peak, where the caseload returns to at least 30% of the previous peak’s caseload”. A total of 286 nurses completed the survey.

The survey questionnaire was available on a data-free website, which meant that the respondents did not need to use their own data to access and complete the questionnaire. Security features of the website included encryption software and password-protected files. Nurses who voluntarily agreed to participate in the study, clicked on an option in the information leaflet indicating that they understood what the study entailed. They were then routed to the survey. Nurses could complete the survey at their workstations or on private devices such as smartphones or tablets. It took approximately 15 minutes for a nurse to complete the questionnaire. There were no direct benefits for participating in the survey, although interested respondents could enter a lucky draw to win one of five food-shopping vouchers to the value of USD 34.53. We also provided contact information for free counselling services for nurses experiencing psychological distress due to COVID-19.

### 2.3. Measures

The questionnaire comprised the following sections: demographic and background information, including sex, age, marital status, job category, place of work, and years of experience; general physical health of nurses; perceived risk factors for COVID-19 infection; infection control—specifically the availability of personal protective equipment (PPE) and the use thereof; as well as two standardized scales to measure PTSD and coping.

Firstly, the Impact of Events Scale-Revised (IES-R), is a standardized measure of post-traumatic stress symptoms, used for recent and specific traumatic events, including COVID-19 [16,17,22,36,37]. It comprises 22 questions, five of which were added to the original Horowitz IES to better capture the Diagnostic and Statistical Manual of Mental Disorders, fourth edition (DSM-IV), criteria for PTSD [38]. Participants are asked to rate the level of distress for each component during the seven days preceding the interview. Items are answered on a five-point Likert scale ranging from 0 (not at all) to 4 (extremely). The scale comprises three subscales: intrusion (intrusive thoughts, nightmares, intrusive feelings and imagery, dissociative-like re-experiencing), avoidance (numbing of responsiveness, avoidance of feelings, situations, and ideas), and hyper arousal (anger, irritability, hypervigilance, difficulty concentrating, heightened startle response), as well as a total subjective stress IES-R score. Scores on the IES-R are categorized as: normal (0–23), mild (24–32), moderate (33–36), and severe psychological impact (>37). On this test, scores higher than 32 are of concern; the higher the score, the greater the likelihood of PTSD and associated health and well-being consequences [39]. A study of the psychometric properties of the IES-R revealed that it appears to be a solid measure of PTSD [40]. We found a 0.9 Cronbach’s alpha coefficient for the overall IES-R, indicating that this scale is reliable for the population we studied (Table 1).

Secondly, the Brief COPE (Coping Orientation to Problems Experienced) Inventory is a 28 item self-report questionnaire designed to measure effective and ineffective ways to cope with a stressful life event. The items are scored on a four-point scale ranging from 1 (I have not been doing this at all) to 4 (I have been doing this a lot). It comprises 14 subscales: Self-distraction, Active coping, Denial, Substance use, Use of emotional support, Use of instrumental support, Behavioral disengagement, Venting, Positive reframing, Planning, Humor, Acceptance, Religion and Self-blame [41]. The Brief COPE scale has been used widely in examining coping behaviors related to traumatic stressors [42], including COVID-19 [21,43,44,45]. Total scores for each subscale are calculated, and higher subscale scores indicate greater perceived use of that coping behavior [42]. Prior research has found that the scales can be grouped into two overarching factors: approach (active coping, planning, positive reframing, acceptance, seeking emotional support, and seeking instrumental support) and avoidant coping styles (self-distraction, denial, venting, substance use, behavioral disengagement, and self-blame). Since the Humour and Religion subscales consist of both adaptive and problematic components, they belong neither to approach nor to avoidance coping [21,43,46]. We found Cronbach’s alpha co-efficient ranging from 0.5 to 0.9 for the various subscales of the Brief COPE. Only two subscales recorded co-efficients below 0.7 (Table 1).

### 2.4. Data Analysis

Data was analyzed in IBM SPSS version 26 (IBM, Armonk, NY, USA). Descriptive statistics were generated yielding frequency counts and percentages for categorical variables and means and standard deviations for continuous variables. Composite scores were calculated for all scales and subscales. Cronbach’s alpha was used to test the internal consistency of the scales and subscales. Binomial logistic regression analysis was used to determine which factors were significantly associated with post-traumatic stress. Scores higher than 32 on the IES-R are indicative of problematic PTSD [39] and was used as a cut-off point for the regression. All assumptions for binomial logistic regression were met. Independent variables included in the model were: age; job category (manager, professional nurse, other nurse); job sector (private, public); treatment of COVID-19 patients (yes, no); level of preparedness to manage COVID-19 patients (fully prepared, not fully prepared), current health compared to before the COVID-19 outbreak (better, unchanged, worse), approach coping, avoidance coping, humor and religion. The open-ended questions were coded and categorized to calculate frequency counts.

## 3. Results

### 3.1. Biographical Characteristics and Overall Health

Most respondents were females (*n* = 248; 86.7%) and the average age was 44.6 years (SD 11.569). Two-thirds were married/in a long-term relationship (*n* = 189; 66.1%) and slightly more than half lived with their partner (*n* = 156; 54.5%) and/or school-going children (*n* = 161; 56.3%). While most respondents indicated that their health was good to excellent (*n* = 244; 85.3%), 30.4% reported that their health had changed for the worse since COVID-19 (*n* = 87). A third of the nurses had been diagnosed with COVID-19 (*n* = 93; 32.5%), with most of these respondents highly concerned about in turn infecting their family (*n* = 74; 79.6%) (see Table 2 for biographical information).

A large majority of respondents reported knowing a colleague who was diagnosed with COVID-19 (*n* = 259; 90.6%), while almost a quarter (*n* = 66; 23.1%) indicated that someone in their household had been diagnosed with COVID-19.

### 3.2. Work Environment

The respondents were classified into three groups: nursing managers (*n* = 49; 17.1%), professional nurses (*n* = 173; 60.5%), and other nurses, including enrolled nurses, assistant nurses and student nurses (*n* = 64; 22.4%). More than two-thirds of respondents worked in the public sector (*n* = 204; 71.3%) which included hospitals, PHC clinics, provincial and district offices, and education facilities, while 28.7% (*n* = 82) worked in private hospitals. They had an average of 19.3 years of experience working as a nurse (SD 12.860).

Three-quarters of the respondents reported screening patients for COVID-19 (*n* = 213, 74.5%), while half of the respondents cared for patients with COVID-19 (*n* = 147; 51.4%). Two-thirds indicated that they had received training on how to care for COVID-19 patients (*n* = 196; 68.5%), while only 38.5% (*n* = 110) felt fully prepared to care for COVID-19 patients. Furthermore, half of the respondents (*n* = 150; 52.4%) were highly concerned about contracting COVID-19 at work. The main risks for contracting COVID-19 at work were indicated as being an HCW (*n* = 219; 76.6%) and the public or patients not adhering to prevention guidelines (*n* = 206; 72%). Lack of staff (*n* = 121; 42.3%); inadequate PPE (*n* = 70; 24.5%); and underlying health conditions (*n* = 68; 23.8%) were also perceived as risk factors for COVID-19 (Table 3).

### 3.3. Post-Traumatic Stress and Coping

Overall, the nurses had a mean score of 31.5 (SD 20.586) on the IES-R. A closer look revealed that 44.4% of nurses scored above 32 on the IES-R, which is indicative of higher levels of PTSD, with 38.8% of the nurses experiencing severe PTSD (Table 4).

From the 14 subscales on the Brief COPE, Acceptance (M 6.1, SD 1.834), Religion (M 6.0, SD 2.0), Planning (M 5.5, SD 1.970), and Active coping (M 5.3, SD 1.936) were rated the highest (Table 5).

Binomial regression (Table 6) was run to ascertain the effects of age; job category (manager, professional nurse, other nurse); job sector (private, public); treatment of COVID-19 patients (yes, no); level of preparedness to manage COVID-19 patients (fully prepared, not fully prepared); adequate PPE (available, not available), current health compared to before the COVID-19 outbreak (better, unchanged, worse); approach coping; avoidance coping; humor; and religion on PTSD. The model was statistically significant, implying that the predictors as a set reliably distinguished between nurses who had normal levels of PTSD and nurses who had higher levels of PTSD, X^2^(13) = 110.309, *p* < 0.005. The model explained 44% (Nagelkerke R^2^) of the variance in the tendency to experience PTSD and correctly classified 76.5% of the cases. After controlling for other variables in the model, three predictor variables were found to be statistically significant (*p* < 0.05)—preparedness to care for COVID-19 patients, avoidance coping strategies and current health compared to pre-COVID-19 health.

More specifically, compared to nurses who experienced better health since COVID-19, nurses who experienced worse health were 4.4 times more likely to have PTSD (OR = 4.435, *p* = 0.018). Compared to nurses who felt fully prepared to deal with COVID-19 patients, nurses who indicated not being fully prepared to care for these patients were 2.3 times more likely to experience PTSD (OR = 2.298, *p* = 0.017). Nurses who were more likely to practice avoidant coping strategies, were also more likely to experience PTSD (OR = 1.206, *p* = 0.000).

### 3.4. Support Needs of Nurses

Two open-ended questions were included to determine the type of support nurses needed from (1) their managers and (2) the provincial health department. Common themes that emerged included: emotional support and acknowledgement, psychological support and debriefing sessions, more nursing personnel, sufficient and appropriate PPE, training and up-to-date information on COVID-19, and financial incentives.

Almost half of the nurses (*n* = 141; 46.5%) indicated that they would like to receive some form of emotional support from their managers, which included: access to counselling/debriefing services (*n* = 47; 33.3%) and words of reassurance, understanding and empathy from managers (*n* = 94; 66.7%), “*Firstly, he or she must show empathy and have time to listen to our concerns. Secondly, he or she must ensure that we get psychological support as part of the Employee Wellness Programme.*” There were also similar requests that the provincial health department should provide emotional support for and acknowledgement of the work done by nurses (*n* = 56; 19.6%) including psychological counselling and debriefing sessions (*n* = 41; 14.3%): “*Organize sessions not once but book for us as much sessions until the psychologist thinks we are fully ready to be in that environment especially if there are after effects it lowers your self-esteem and emotionally.*”

There were also requests for managers (*n* = 20; 6.9%) and the provincial health department (*n* = 30; 10.5%) to recruit more staff and to use staff more efficiently. Rotation was suggested as a means to prevent prolonged exposure to COVID-19 patients: “*Rotation when it comes to nursing COVID patients* […] *just because I’m a student it does not mean that I won’t get it or I don’t have a family to go home to or I’m not scared of the virus. We all are* […] *it’s affecting all of us and we should all be working together to fight against it. It shouldn’t only be the nurses’ responsibility alone. It’s a pandemic which affects all of us and has the same effects on all of us regardless of age or appearance. Managers should please allow all of us nurses to nurse COVID patients. We all took an oath.*”

Nurses also indicated that they should be provided with sufficient and good quality PPE (managers *n* = 26; 9.6%; provincial health department *n* = 28; 9.8%). It was suggested that “*All PPE should be procured from province, institutions SCM* [supply chain management] *because they are not nurses, don’t do their work and at the end we work without proper PPE* e.g., *plastic aprons, they are now telling us that there is no money.*”

There were also requests for more training and up-to-date information on COVID-19 (managers *n* = 18; 6.3%; provincial health department *n* = 13; 4.6%). Nurses who tested positive for COVID-19 needed assistance with medical care: “*Give us the medication course we don’t have money to buy medication* […] *medical aid didn’t pay for it*” and emotional support: *“Ensuring that they receive counseling not only when they are sick, but throughout this period, it should be done routinely and free.*” Concerns were also raised about sick leave when testing positive for COVID-19: “*Special leave is to be granted if you tested positive. Currently they deduct it from your sick leave.*”

Finally, almost a quarter of the nurses (*n* = 66; 23.1%) indicated that the provincial health department should provide financial support in the form of danger pay or COVID-19 bonuses: “[…] *incentives, due salary increase plus 5% for nurses as nurses were at the physical forefront for this pandemic for even longer periods than even other* [HCWs] […] *I am not discrediting others’ roles*”; and “*Also give us Risk Allowance as we didn’t expect this monster and we risked our lives to take care of patients in this difficult times.*”

## 4. Discussion

While numerous studies report on the psychosocial well-being of HCWs during COVID-19 [14,15,16,17,18,19,22,29,36,37,43,44], ours is one of the first to examine the influence of the pandemic on the post-traumatic stress and coping strategies of nurses in South Africa—one of the most unequal countries in the world [47]. Given the myriad of stressors already faced by nurses in the country amidst an ailing healthcare system, it is paramount to identify coping strategies and needs of nurses in order to inform support interventions for the mental healthcare of nurses during times of crisis.

Data collection occurred during the second wave of the pandemic in the country. We found that 44% of nurses screened positive for PTSD, which was higher than recorded in a study conducted earlier in the year where only approximately 20% of HCWs were reported to be severely distressed [28]. Our overall mean score of PTSD was similar to that reported for the general population in China in January to March 2020 [48]. While a Spanish study [37] reported that 56.6% of HCWs presented with high levels of PTSD, they used a lower score of 20 on the IES-R to define PTSD, our study used a cut-off score of 32 as suggested by Creamer et al. [39]. In general, when compared to international studies among HCWs, our study had a higher percentage of nurses with clinically concerning levels of PTSD. Lai et al. [17] found clinically concerning levels of PTSD among 35% of Chinese nurses. Ali et al. [22] reported in their study among Irish HCWs that 41% experienced higher levels of PTSD, although they also used a lower cut-off point (24 vs. 32) to categorize higher PTSD. Chew et al. [36] found that that HCWs in Singapore and India displayed lower levels of PTSD, with 7.4% of the study cohort screening positive for clinical concerning PTSD (also using 24 as a cut-off point). Tan et al. [49] found similar levels of PTSD amongst HCWs in Singapore (also using 24 as a cut-off point). These comparisons should be interpreted with caution, as the studies were conducted at different time periods of the pandemic in the respective countries. Despite this, it is evident that South African nurses were experiencing comparatively higher levels of post-traumatic stress than their counterparts in other countries.

Higher levels of post-traumatic stress among all categories of South African nurses (i.e., there was no significant difference in post-traumatic stress levels between managers, professional nurses and other nurses) likely also relate to the state of the healthcare system in the country. For example, South Africa has 1.3 nurses/midwives per 1000 population compared to 3.8 globally [50]. According to estimates from the International Council of Nurses [51], on average 7% of all COVID-19 cases worldwide occurred among HCWs. Currently in South Africa, this translates to 111,175 HCWs infected with COVID-19 thus far [52]. In this regard, our respondents called for more nursing staff to assist with the management of COVID-19 patient care, as well as the rotation of staff to prevent prolonged exposure in COVID-19 wards. This places an enormous burden on an already constrained system fraught with disparities and corruption. Notable in this regard is the reported tender fraud related to the procurement of PPE for COVID-19. A special investigation into this, referred to the fraud as “*a flagrant and wanton disregard of the applicable law, policies and procedures*” [53]. Unfortunately, HCWs suffer the immediate consequences of such actions. The Director General of the World Health Organization noted “*Any type of corruption is unacceptable* […] *However, corruption related to PPE* […] *for me it’s actually murder. Because if HCWs work without PPE, we’re risking their lives. And that also risks the lives of the people they serve*” [54]. Univariate analysis found that lack of PPE was associated with increased post-traumatic stress in our study and there were numerous requests for more PPE. Nurses in other countries, particularly less prosperous countries with poor health systems, have also reported a lack of PPE during the first wave of COVID-19 [55,56].

Self-reported risk for contracting COVID-19 mainly centered on being an HCW and patients and the public not adhering to COVID-19 infection prevention and control guidelines. Lai et al. [17] also reported that frontline HCWs in China were at higher risk for PTSD. Regression analysis did not, however, find that Free State nurses who provided care for patients with COVID-19 experienced higher levels of PTSD. A potential explanation for this finding is the PHC approach and scarcity of medical doctors—0.9 per 1000 population [57] in South Africa, which positions nurses as the first port of call for the majority of the population. It is also interesting to note that there were no significant differences in post-traumatic stress levels when comparing nurses who work both in the private and public sector, which may in part relate to the fact that, possibly for the first time in the country, we saw strong collaboration between the private and public sector in managing COVID-19 [58,59]. This suggests a shared responsibility and a notion of working for the common good among nurses in both sectors. It also speaks to universal elements that impact the psychosocial well-being of all HCWs. This was clear from our regression analysis which found that unpreparedness to manage COVID-19 patients, poorer health, and avoidant coping strategies were associated with PTSD.

While the unpreparedness to care for COVID-19 patients was associated with PTSD, this did not appear to relate to training per se: firstly, a high percentage of nurses reported receiving COVID-19 related training. Secondly, there were not many requests for training on COVID-19. However, there were numerous requests for emotional and psychological support for nurses both from immediate managers and the provincial health department. A systematic review of post-traumatic stress symptoms in HCWs during COVID-19 highlighted a lack of social support as a predictor of post-traumatic stress [55]. Similarly, Du et al. [14] found that the lack of psychological preparedness to deal with COVID-19 was associated with elevated depressive and anxiety symptoms in HCWs. When one considers how COVID-19 has altered the social and working environment (i.e., social distancing, wearing of masks, isolation, lockdowns, etc.) the added need for emotional support, even just in the form of encouraging words and acknowledgment for work done, is understandable. Several authors have alluded to the importance of providing HCWs with support to ensure their psychological well-being [2,14,15,16,22,25,36,44,55].

We found an association between nurses who viewed their current health as being worse since COVID-19 and PTSD. This is supported by other research which reported that underlying illnesses are a risk factor for poor psychological health during COVID-19 [24]. Descriptive analysis points to almost a quarter of our nurses indicating that an underlying illness placed them at risk for COVID-19. More specifically, almost a quarter of our nurses indicated having hypertension which is associated with severe COVID-19 and fatal outcomes [60].

Finally, the research found an association between avoidant coping styles and PTSD. Avoidant coping styles include self-distraction, denial, venting, substance use, behavioral disengagement, and self-blame. According to the literature, individuals opt for dysfunctional coping strategies when faced with an uncontrollable event (e.g., the COVID-19 pandemic). When this occurs, the focus is on coping with the problem and not on dealing with emotions [61], and is evidenced in our study, as well as similar research by McFadden et al. [44] and Canestrari et al. [43] who reported that avoidant coping behaviors were associated with lower levels of well-being among UK health and social care workers and higher levels of stress in Italian HCWs, respectively. These findings were corroborated by d’Ettorre et al. [55] in a systematic review, which found that a passive coping style was a risk factor for post-traumatic stress during COVID-19. This is in line with our finding that many nurses require emotional support (i.e., to cope with emotions and not avoid the problem), words of reassurance, understanding and empathy from their managers.

Based on the findings of this study, the following areas require attention to address increased levels of post-traumatic stress amongst all categories of nurses working both in the public and private healthcare sectors in South Africa: emotional support from managers; psychological interventions and debriefing sessions that focus on positive coping strategies to actively address stressful events; and the provision of sufficient and appropriate PPE. Nurses who are emotionally and physically equipped to manage COVID-19 at work, at home and in the broader community, will have a better chance at dealing with stressful events. These are not new recommendations, but have already been made in relation to coping with other pandemics, for example the 2014 Ebola outbreak in West Africa [3], the 2006 SARS outbreak in Canada [5], and the 2005 SARS outbreak in Taiwan [62], where the importance of appropriate support and available PPE were repeatedly highlighted.

As with all research, our study has limitations. Firstly, we used an online survey to collect data, which could have been accessed by anyone who had seen the advertisement for the study. We attempted to mitigate against this by commencing with a filter question, where only respondents who indicated that they were a nurse, could continue with the survey. In addition, while the study was advertised on social media, our main recruitment drive was to work through public and private healthcare managers to advertise the study. Secondly, the sample size was small, and a larger sample is required to verify the results. Thirdly, the cross-sectional nature of the study does not allow interpretation for causality and due to changes in post-traumatic stress, dynamic observation would add value to such research. Fourthly, as this was a self-administered online questionnaire, we were cognizant of the length of the questionnaire and the time it would take to complete. As such, we were limited in the number of questions that we could ask and potentially excluded issues that could also have contributed to increased levels of PTSD, for example education level, salary level and shift working status.

## 5. Conclusions

This research among public and private sector nurses in the Free State found that more than four in every 10 nurses screened positive for PTSD. Self-reported risk for contracting COVID-19 mainly centered on being an HCW and other persons’ non-adherence to COVID-19 infection prevention guidelines. Unpreparedness to manage COVID-19 patients, poorer health, and avoidant coping were associated with PTSD. Nurses voiced a need for emotional support and empathy from their managers. Emotional, psychological, and debriefing intervention sessions that focus on positive coping strategies to actively address stress; positive and open communication between managers and their subordinates; psychological support through the Employee Assistance Programme and occupational health units; referral for more intensive psychotherapy where necessary; regular debriefing sessions where nurses can share their experiences; positive messaging from the provincial and national health departments; and an uninterrupted supply of quality PPE are recommended.

## Figures and Tables

**Table 1 ijerph-18-07919-t001:** Reliability of the scales.

Scales	No. of Items	Cronbach’s Alpha Coefficient
**Impact of Life Events Scale Revised** (**IES-R**)	22	0.9
Intrusion	8	0.9
Avoidance	8	0.9
Hyper arousal	6	0.9
**Brief COPE**	28	
**Approach Coping**	12	0.9
Active coping	2	0.7
Use of emotional support	2	0.8
Use of instrumental support	2	0.7
Positive reframing	2	0.7
Planning	2	0.7
Acceptance	2	0.8
**Avoidant Coping**	12	0.8
Self-distraction	2	0.6
Denial	2	0.7
Venting	2	0.5
Substance use	2	0.9
Behavioral disengagement	2	0.8
Self-blame	2	0.8
**Humor**	2	0.8
**Religion**	2	0.8

**Table 2 ijerph-18-07919-t002:** Biographic characteristics and overall health.

Characteristics	*n*	%
**Sex**		
Male	37	12.9
Female	248	86.7
Prefer not to answer	1	0.3
**Age** (mean, standard deviation—SD)	44.6	11.569
**Married/in a long-term relationship**	189	66.1
**Living arrangements**		
Live with partner	156	54.5
Live with school-going children	161	56.3
**People in household** (mean, SD)	3.29	1.604
**Overall health**		
Excellent	55	19.2
Very good	96	33.6
Good	93	32.5
Fair	37	12.9
Poor	5	1.7
**Current health compared to prior to COVID-19 outbreak**		
Better	27	9.4
Unchanged	172	60.1
Worse	87	30.4
**Co-morbidities**		
Hypertension	65	22.7
Cholesterol	30	10.5
Diabetes mellitus	22	7.7
Asthma	18	6.3
Ischemic heart disease	4	1.4
**COVID-19 diagnosed**	93	32.5
Self-isolated	85	91.4
Isolated in a designated facility	5	5.4
Did not isolate	3	3.2
**Concerned about transmitting COVID-19 to family**		
Not concerned at all	3	3.2
Moderately concerned	16	17.2
Highly concerned	74	79.6

**Table 3 ijerph-18-07919-t003:** COVID-19 risk perception.

Characteristics	*n*	%
**Preparedness to care for COVID-19 patients**		
Unprepared	43	15.0
Somewhat unprepared	29	10.1
Unsure	40	14.0
Somewhat prepared	64	22.4
Prepared	110	38.5
**Concerned about contracting COVID-19 at work**		
Not concerned at all	13	4.5
Moderately concerned	123	43.0
Highly concerned	150	52.4
**Risks for COVID-19**		
Profession as HCW	219	76.6
General public do not adhere to prevention guidelines	206	72.0
Inadequate personal protective equipment (PPE)	70	24.5
Workplace not equipped for COVID-19	59	20.6
Underlying health conditions	68	23.8
Lack of staff	121	42.3
Long working hours	62	21.7
Public transport to come to work	42	14.7
Family members do not adhere to prevention guidelines	13	4.5

**Table 4 ijerph-18-07919-t004:** Levels of post-traumatic stress (IES-R).

Impact of Events Scale-Revised (IES-R)	*n*	%
Normal	108	37.8
Mild	51	17.8
Moderate	16	5.6
Severe	111	38.8

**Table 5 ijerph-18-07919-t005:** Coping strategies (Brief COPE).

Subscales	Range	Mean	SD
**Approach Coping**	12–48	31.8	9.051
Active coping	2–8	5.3	1.936
Use of emotional support	2–8	5.0	1.978
Use of instrumental support	2–8	4.8	1.957
Positive reframing	2–8	5.0	1.930
Planning	2–8	5.5	1.970
Acceptance	2–8	6.1	1.834
**Avoidant Coping**	12–48	21.8	6.920
Self-distraction	2–8	4.9	1.935
Denial	2–8	3.2	1.700
Venting	2–8	4.2	1.668
Substance use	2–8	2.8	1.537
Behavioral disengagement	2–8	3.3	1.717
Self-blame	2–8	3.3	1.800
**Humor**	2–8	2.7	1.845
**Religion**	2–8	6.0	2.000

**Table 6 ijerph-18-07919-t006:** Factors associated with higher levels of post-traumatic stress.

Variables	*n* (%)	Unadjusted Odds Ratio (95% CI)	Adjusted Odds Ratio (95% CI)
**Age** (mean, SD)	44.6 (11.569)	0.983 (0.963–1.003)	0.996 (0.965–1.028)
**Job category**			
Manager (ref)	49 (17.1)	1	1
Professional nurse	173 (60.5)	0.939 (0.496–1.778)	0.649 (0.277–1.519)
Other nurse	64 (22.4)	1.083 (0.513–2.285)	0.671 (0.229–1.970)
**Job sector**			
Private (ref)	82 (71.3)	1	1
Public	204 (28.7)	0.845 (0.503–1.420)	0.767 (0.379–1.554)
**Treat COVID-19**			
Do not treat patients (ref)	139 (48.6)	1	1
Treat patients	147 (51.4)	0.722 (0.452–1.154)	1.510 (0.782–2.916)
**Level of preparedness to manage patients with COVID-19**			
Fully prepared (ref)	110 (38.5)	1	1
Not fully prepared	176 (61.5)	2.505 (1.517–4.134)	2.298 (1.163–4.540)
**PPE** *			
Adequate PPE (ref)	216 (75.5)	1	
Inadequate PPE	70 (24.5)	2.137 (1.235–3.697)	1.306 (0.625–2.728)
**Current health compared to prior to COVID-19 outbreak**			
Better (ref)	27 (9.4)	1	1
Unchanged	172 (60.1)	1.272 (0.526–3.078)	1.959 (0.617–6.225)
Worse	87 (30.4)	5.004 (1.954–12.819)	4.435 (1.286–15.292)
**Approach Coping** (mean, SD)	31.8(9.051)	1.074 (1.043–1.106)	1.023 (0.972–1.076)
**Avoidant Coping** (mean, SD)	21.8 (6.920)	1.238 (1.172–1.307)	1.206 (1.131–1.286)
**Humor** (mean, SD)	2.7 (1.845)	1.246 (1.093–1.420)	1.028 (0.862–1.226)
**Religion** (mean, SD)	6.0 (1.989)	1.100 (0.976–1.240)	0.973 (0.831–1.140)

* Personal Protective Equipment (PPE).

## Data Availability

Data supporting reported results can be requested from the first author.

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
