# Peer review of "Post-Traumatic Stress and Coping Strategies of South African Nurses during the Second Wave of the COVID-19 Pandemic"

_ijerph, 2021, doi:10.3390/ijerph18157919_

Round 1
Reviewer 1 Report
Dear authors,
Your manuscript is interesting but I need you to answer some questions:
MATERIALS AND METHODS
Design and Setting:
- The authors have not specified how many health centers there are in the province. Nor have they specified how many centers are public and how many are private.
Sample and Data Collection:
- How was the sample chosen? The authors must specify it.
- The authors must include the response rate of the participants in the study.
RESULTS
- The authors have not given the "results" separated by categories and health centers. The roles and responsibilities of a nurse are not the same as a nursing assistant. Furthermore, healthcare pressure is not the same in public and private centers. Therefore, the "results" may be biased.
REFERENCES
- Some references have bugs. The authors should review this section.
Author Response
Dear Reviewer
Thank you for your valuable comments. Please see our response to your feedback in the attached file.
Kind regards
Michelle Engelbrecht

Reviewer 2 Report
This study is a cross-sectional survey study related to the mental health of South African nurses in the era of COVID-19. Globally, the pandemic is having a major impact on mental health, and as the authors point out, this study are considered to have public health relevance, given the existing medical resource vulnerabilities in South Africa. In addition, the authors' discussion was in-depth.
I would like to add some comments to improve the quality of this paper.
- Readers are probably not familiar with ‘the second wave of COVID-19 in South Africa.’ The kind explanation of the authors can facilitate the reader's understanding.
- The importance of post-traumatic stress seems to be under-emphasized in the Introduction. Among the many aspects of mental health, why should post-traumatic stress be observed? What significance does it have?
- It would be nice to add more detail about the author's survey procedure. What is the “data-free website” the authors used as their survey? Did participants receive any potential benefits or disadvantages related to this survey?
- Among the abbreviations used in the manuscript, undefined expressions are observed. For example, authors could define an acronym for Brief COPE. Also, COPE and Cope are both used through the manuasciprt. I think it should be unified.
- I think it would be good to explain the abbreviations used in the table in footnotes below the table. (e.g. SD, COVID-19, HCW, PPE, etc.)
- Is “Title 3” a typo in Table 1?
- In Table 1, Humour and Religion seem to need bolding.
- I think the limitations of the authors' research can be further reinforced. Were the participants' education level, salary level, and shift working status investigated? Could it potentially affect the outcomes of interest in this study? Was it discussed in the Discussion section?
- What is ‘90.6?%’ on page 6?
- In Table 6, it seems more appropriate to change the nurse to other nurse among the job categories. Does PPP stand for PPE? ‘Adeque PPE’ and ‘Better’ should be marked with (ref) added. ‘Not fully prepared’ requires center alignment.
Author Response

(The authors gave the same response as above.)

Round 2
Reviewer 1 Report
Dear authors,
Thanks for your reply. The explanations of the authors are satisfactory. The paper has greatly improved its quality.
Congratulations on your work.
Best regards